


# Will UK peatland restoration reduce dissolved organic matter concentrations in upland drinking water supplies?

Jennifer Williamson[1*], Chris Evans[1], Bryan Spears[2], Amy Pickard[2], Pippa J. Chapman[3], Heidrun Feuchtmayr[4], Fraser Leith[5], Don Monteith[4]

[1]UK Centre for Ecology & Hydrology, Environment Centre Wales, Deiniol Road, Bangor, Gwynedd, LL57 2UW

[2]UK Centre for Ecology and Hydrology, Bush Estate, Penicuik, Midlothian, EH26 0QB

[3] School of Geography, Faculty of Environment, University of Leeds, Leeds, LS2 9JT

[4]UK Centre for Ecology & Hydrology, Lancaster Environment Centre, Library Avenue, Bailrigg, Lancaster, LA1 4AP

[5]Scottish Water, 6 Castle Drive, Dunfermline, KY11 8GG

[*]*Corresponding Author* (jwl@ceh.ac.uk)

**Abstract**

Rising dissolved organic matter (DOM) concentrations, and associated increases in water colour, have posed a potential problem for the UK water industry since the phenomenon was first reported in the early 1990s. Elevated DOM concentrations in raw water are of particular concern in upland catchments dominated by
organic soils, where DOM production tends to be highest. In recent years, water companies have considered the capacity for catchment interventions to improve raw water quality at source, relieving the need for costly and complex engineering solutions in treatment works, but there is considerable uncertainty around the effectiveness of these measures. One of the primary evidence gaps is the extent to which catchment management is capable of influencing DOM concentrations at the point of abstraction, field studies rarely extending beyond sub-catchment
or stream scale. Our review of the published evidence suggests that catchment management could make a contribution to mitigating recent DOM increases in some circumstances, particularly where plantation forestry has been grown on peat, and where control of nutrients in runoff could reduce in-reservoir DOM production. Evidence for the efficacy of most other measures that target reductions in DOM loading for catchment to reservoir remains insufficient to support wider scale application. Collectively, these measures have the potential
to reduce DOM concentrations in drinking water reservoirs but they must be selected on a site-specific basis, where the scale, effect size and duration of the catchment intervention are considered in relation to both the treatment capacity of the works and future projected DOM trends.





## 1. Introduction

### 1.1 What is dissolved organic matter (DOM)?

Dissolved organic matter (DOM) concentrations in UK surface waters have been increasing since as early as the 1980s, according to water colour monitoring records maintained by the UK water industry (e.g. Naden and McDonald, 1989;Watts et al., 2001). Systematic scientific monitoring of DOM in UK upland waters began in the 1980s at research sites in mid-Wales (Plynlimon and Beddgelert catchments) and a series of lochs and streams in Scotland monitored by the Freshwater Fisheries Laboratory in Pitlochry. In both regions, consistent patterns of decadal-scale increases in DOM were becoming evident by the mid-1990s (e.g. Harriman et al., 2001;Robson and Neal, 1996). By the early 2000's it was evident that these trends extended across the UK (Freeman et al., 2001;Worrall et al., 2004) and subsequently DOC increases have emerged as an international phenomenon affecting surface waters across large areas of Europe and North America (Monteith et al., 2007) Records, such as those of the UK Upland Waters Monitoring Network, demonstrate these have continued to the present (e.g. Battarbee et al., 2014).

DOM comprises a spectrum of organic molecules from colourless simple sugars to highly-complex humic structures containing chromophores that strongly absorb light in the UV and visible regions of the spectrum. The elemental composition of DOM is dominated by carbon, hydrogen, oxygen but also includes a range of other macro-nutrients, including nitrogen and phosphorus, and micronutrients. The light absorbing properties of humic compounds impart a yellow-brown colouration to water – referred to as water colour. Due to the diversity of DOM molecular composition, water colour metrics provide an indirect measure of total DOM. However, waters draining organic humic soils show a strong positive relationship between water colour and DOM, and absorbance at ~400 nm is widely applied as a DOM metric for such systems. A more direct measure of DOM is provided by the measurement of all the organic carbon dissolved in the water (i.e. Dissolved Organic Carbon (DOC)), which comprises an approximately fixed proportion of the total DOM. DOM can be broadly characterised as hydrophobic or hydrophilic, with the former comprising high molecular weight molecules including lignins and humic acids and the latter primarily lower molecular weight aliphatic molecules. Hydrophobic DOM is more easily removed by conventional coagulation and filtration during drinking water treatment due to the presence of charged functional groups (Matilainen et al., 2010). The hydrophobicity of the raw water entering a treatment works can be estimated using Specific UV Absorbance measurements at 254 nm ($SUVA_{254}$), calculated using the total DOC concentration and absorbance at 254 nm (Weishaar et al., 2003). Values greater than 4 indicate that DOM is primarily hydrophobic, while values less than 2 show the DOM is primarily hydrophilic and will not be effectively removed using conventional coagulation and filtration alone (Matilainen et al., 2010). Hydrophobic DOM is primarily allochthonous in origin, particularly in waters draining organic soils and tends to be photoreactive and biologically recalcitrant; whereas the hardest to remove through conventional treatment, most hydrophilic, DOM is mostly produced within the waterbody through phytoplankton activity (Imai et al., 2002) and is biologically labile but photostable (Berggren and del Giorgio, 2015;Berggren et al., 2018).**1.2 DOM management by the water industry**

Within the UK the need for DOM removal is driven by regulatory measures, including the requirements that water colour reaching customers' taps is less than 20 mg l$^{-1}$ Pt/Co (hazen); total tri-halomethane (THM) concentrations are less than 100 µg l$^{-1}$; and E. coli and Enterococci counts are 0 per 100 ml (DWI, 2014;The Public Water Supplies (Scotland) Regulations, 2014). These requirements have been translated into Scottish and English and Welsh law from the European Drinking Water Directive (Drinking Water Directive (98/83/EC)). Although consumption of DOM in drinking water is not directly harmful to people, perceptibly coloured water reduces customer satisfaction and can be indicative of further problems. Indirectly, elevated DOM concentrations have implications for human health due to their potential influence on treatment processes and the production of carcinogenic disinfectant by-products (DBPs) such as THMs.

Chlorination, a standard disinfection process in most UK WTWs, leaves free chlorine in the water supply as a residual disinfectant. Free chlorine reacts with DOM remaining in the water supply to form DBPs including THMs, which are regulated by the Drinking Water Inspectorate (DWI) due to their potential carcinogenic properties. Chloramination, the treatment of drinking water with chlorine and ammonia to form chloramine, has been used at WTWs as a method of reducing THM formation. However, it has been found that chloramination promotes the formation of nitrogenous DBPs (e.g. Bond et al., 2011;Lavonen et al., 2013), which are more carcinogenic than THMs (Ding and Chu, 2017) and are likely to be regulated in the future. DOM may hamper the



efficacy of chlorine as a disinfectant while simultaneously acting as a substrate for bacterial regrowth, thus increasing the risk of regulatory failure from bacterial contamination and the subsequent loss of customer trust.

Higher concentrations of DOM in raw water therefore necessitate a greater amount of treatment to prevent the issues listed above. This may include larger coagulant dosages, shorter filter run times, longer and more frequent cleaning of filtration units, higher energy costs, higher sludge removal costs and an increase in direct and indirect (energy-related) greenhouse gas (GHG) emissions from the treatment process (Jones et al., 2016). Overall, the cost of DOM removal in UK water supplies is estimated to be hundreds of millions of pounds, and has risen sharply in recent years as a direct consequence of rising DOM concentrations. Major additional costs are incurred
where capital investment is needed to upgrade treatment infrastructure that is no longer able to deal with current concentrations. The water industry therefore requires greater understanding of how and where DOM concentrations and quality/treatability are likely to change in future to inform adaptation and mitigation strategies.

### 1.3 What is driving DOM increases?

Since the early 2000's a number of hypotheses have been advanced to explain regional scale increases in DOM
concentrations. These initially focussed on increasing temperatures (e.g. Freeman et al., 2001), drought-rewet cycles (e.g. Watts et al., 2001), and increasing atmospheric carbon dioxide concentrations (Freeman et al., 2004), before negative correlations with indicators of acid deposition, such as sulphate concentration, became increasingly apparent in the USA (Stoddard et al., 2003) and UK (Evans et al., 2005). A regional study of DOC trends (Monteith et al., 2007) demonstrated consistent significant negative relationships between rates of change
in acid anion concentrations (sulphate & chloride) and rates of change in DOC, and that the effect was more marked for waters with lower concentrations of calcium and magnesium (i.e. base cations). Hence, sites with soils that were least able to buffer the effects of deposited acidity were the most responsive. The links with changes in atmospheric deposition have since been supported by studies of soil cores (Clark et al., 2011) and field experiments (Evans et al., 2012;Ekström et al., 2015). Hruska et al. (2009) demonstrated that ionic strength (a
measure of the electric charge produced by ions in water) is a particularly effective chemical predictor of change in DOC. A reduction in the deposition of acid anions from the atmosphere reduces both the acidity of soil and the ionic strength of soil water, and together these processes appear to increase the solubility of soil organic matter and hence the concentration of DOM draining from organic rich upland soils.

DOM concentration in soil solution and surface waters is also known to respond positively to variation in
temperature (e.g. Vance and David, 1991), while shifts from vertical to more lateral routing of flow paths during periods of heavy rain have also been found to increase concentrations in some circumstances (e.g. Austnes et al., 2010), with increases in DOC concentration being primarily driven by the increase in water table at the event scale (Rosset et al., 2019). Shifts in stream DOC character, and hence treatability, are also possible following changes in flow path routing as a result of DOC inputs from different source pools (Hood et al., 2006). Long-term increases
in DOC in southern Sweden have been linked to the combination of decreasing sulphate deposition and a multi-decadal increase in precipitation and consequently river flow (Erlandsson et al., 2008). Future climate change, particularly in relation to increasing temperatures and a change in total rainfall and an increase in the intensity of storm events (Met Office, 2019) is therefore likely to influence future DOM trajectories and has the potential to become the dominant driver as atmospheric pollutant deposition declines toward pre-industrial levels.

### 1.4 Impacts of catchment management

Upland areas provide around 70% of the UK's drinking water, a large proportion of which derives from organic soils such as peatlands (Xu et al., 2018). The catchments of many UK upland water sources used for water abstraction have been subjected to drainage, plantation forestry and burn-management and the perception that these activities have caused, or at least exacerbated rising DOM trends, has often formed part of the rationale for
restoring degraded peatlands (e.g. United Utilities, 2018). Importantly, restoration also delivers a range of other benefits including reductions in bare ground and therefore erosion, reductions in peat oxidation and associated $CO_2$ emissions, natural flood management, improvements in habitat condition and biodiversity (Curtis et al., 2014), and enhancements of amenity value (for example the restoration work carried out on the highly eroded Kinder Plateau in the Southern Pennines, England by Moors for the Future). To date, evidence to support the
contention that a) local catchment degradation has made a significant contribution to long-term DOM increases, and b) that peatland restoration can help to reduce DOM concentrations at the point of abstraction for drinking water treatment, has been limited and awaits careful evaluation. Despite this, in recent years a number of water companies, with the encouragement of the UK industry regulator, have begun prioritising investment in peatland





restoration within drinking water catchments over more energy intensive and costly treatment plant upgrades, with the aim of improving water quality at source. The potential role of reservoir processing on both DOM concentrations and DOM treatability has received relatively little attention within the water industry, yet it may be a major factor as to why catchment management based changes seen at a plot scale are not seen at the point of abstraction.

The aim of this review is to bring together the information available in the published literature that contributes to our understanding of how catchment management could help mitigate recent and possible future increases in DOM concentrations in upland water bodies within peat dominated catchments. The main focus is on evidence for the effects of the following on DOC concentrations and treatability: drainage ditch blocking, revegetation of bare peat and changing the dominant vegetation species present, the effects of plantation forestry and forest management and managed burning on peat. We go on to examine the relatively unstudied potential role of in-reservoir processing on DOM levels in raw water supplies, before summarising current knowledge gaps and suggesting priorities for future work.

## 2. Current evidence for the efficacy of catchment management approaches in the reduction of DOM

### 2.1. Ditch blocking

Peatland areas in several parts of the UK uplands, including those serving as drinking water catchments, were extensively drained in the post-war period between the 1950s and 1980s in an attempt to improve agricultural productivity; over 1.5 million ha are estimated to have been drained in total (Parry et al., 2014). Large areas of blanket bog were also drained to support the establishment of conifer plantations, most notably in northern Scotland. In theory, resulting reductions in water tables, loss of peat forming plant species, and the cracking of the peat surface, could have exposed previously permanently saturated organic matter to oxidative processes, making it more vulnerable to erosion and, potentially, dissolution into DOM. More recently, extensive efforts have been made by the water industry and organisations concerned with peatland conservation to block ditches in an attempt to restore the hydrological, biogeochemical and ecological functions of these landscapes (Figure 1).

A number of studies (n=4) have reported significant changes in DOC concentrations within peat soil pore water following ditch blocking. Most indicate decreases, with a cross-study average 37% reduction (Haapalehto et al., 2014;Holl et al., 2009;Menberu et al., 2017;Wallage et al., 2006). While these results are persuasive, they do not necessarily imply that effects will be translated through to surface waters and ultimately to the point of abstraction.

At the ditch scale, results are more variable and not consistent with those for pore waters (Table 1). The studies we have identified (n=10) show a mean 10% increase in DOC concentrations following ditch blocking, although this figure is skewed by the large increases reported by Worrall et al. (2007b) and Haapalehto et al. (2014), and the median change is 0. Importantly, no significant change in DOC concentration was reported in half of these studies (Armstrong et al., 2010;Evans et al., 2018;Gibson et al., 2009;O'Brien et al., 2008;Wilson et al., 2011). Differences in apparent effect size may be related to experimental design, including whether the work included a simultaneous control and the time period over which post-restoration monitoring was carried out.

Despite the largely equivocal evidence for ditch-scale effects of blocking on DOC concentrations, there is a tendency for studies that have simultaneously considered hydrological impacts of ditch blocking to report reductions in DOC fluxes, with an average 24% reduction across all reported studies. However, interpretation of these results is not straightforward. For example, Wilson et al. (2011) reported a greater than tenfold reduction in DOC flux from blocked versus unblocked catchments, despite DOC concentrations remaining almost identical. Although the concept that ditch-blocking 'holds water up on the hill' has gained considerable traction, blanket bogs have limited water storage capacity, and cannot act as ever-increasing stores for precipitation over many years. In a previous review of drainage impacts on DOC losses, Evans et al. (2016) considered this study to be methodologically unreliable due to the physically implausible changes in flow reported, and excluded it from their data synthesis. In a recent study of DOC responses to blanket bog re-wetting in North Wales, Holden et al. (2017) showed that damming of drainage ditches did reduce discharge along the original ditch lines, but that most or all of the displaced flow was instead leaving the peatland via overland flow or near-surface through-flow. Evans et al. (2018) subsequently showed that DOC concentrations in water displaced along these pathways were approximately the same as those in water travelling along the ditches, with the result that ditch-blocking was not found to have any clear effect on either DOC concentrations or fluxes at the catchment scale. In summary, any


study reporting large reductions in DOC flux without commensurate reductions in DOC concentration should be treated with a degree of caution.

A number of studies (n=9) have assessed the potential impact of this restoration method on DOM quality and hence the ease of treatability within a conventional water treatment works. These are reviewed within Peacock et al. (2018). They found that the majority of studies at UK and continental European ditch blocking locations, along

with results from their experimental work, showed little effect of ditch blocking on DOM treatability as measured by commonly reported metrics such as SUVA, E2:E3 ratios (ratio of light absorbance at 250 and 365 nm) and E4:E6 ratios (ratio of light absorbance at 465 and 665 nm) (Peacock et al., 2018;Strack et al., 2015;Gough et al., 2016;Glatzel et al., 2003;Lundin et al., 2017). On the other hand, Evans et al. (2014) did find that the radiocarbon ($^{14}$C) content of DOC did vary with peatland drainage and restoration, with intact peatlands tending to export

DOC with a high $^{14}$C content (derived from plants and litter), drained peatlands exporting $^{14}$C-depleted DOC (indicative of the release of carbon by deeper peat) and restored peatlands showing a partial reversal of this response. This suggests that the source of DOC, if not its composition, is affected by drainage and restoration. However observed $^{14}$C changes were larger in the more readily drained peatlands of Scandinavia, continental Europe and Southeast Asia than in the blanket bogs of the UK, which are hydrologically more resistant to drainage.

More broadly, while the evidence from pore water studies hints at possibly significant effects of ditch-blocking at that scale, evidence that such activities have successfully influenced DOM concentrations, and indeed fluxes, at a catchment scale is lacking. It is important to note, however, that catchment-scale studies are hugely challenging logistically and financially to design and maintain and are currently very rare. We are aware of other ongoing, and currently un-published, studies that have the potential to shed more light on this issue in future.

**2.2. Re-vegetation of bare peat**

Exposure of bare peat following anthropogenic disturbance has been an extensive problem in a number of UK peatland regions, most notably in the Peak District. The subsequent erosion of the peat has caused significant problems for the water industry because of the high particulate loads from the catchment to the downstream reservoirs. There has been considerable interest and activity in recent years focussed on restoring vegetation cover

in these areas, with perceived multiple benefits argued to include reductions in water colour and improvements in terrestrial and aquatic biodiversity.

Published research on the impacts of revegetation of peatland areas on DOM is limited, but Qassim et al. (2014) found that pore water DOM concentrations were higher in revegetated sites compared to bare peat areas and vegetated controls over a five year period. The initial revegetation mix in this work was a nurse crop of *Agrostis*

sp., *Deschampsia flexuosa* and *Festuca* sp. in combination with additions of lime and fertiliser to improve grass growth. Heather brash was applied to stabilise the surface and provide a seed source of peatland species. Lime inputs may have increased DOM solubility through a reduction in acidity of the peat (Evans et al., 2012), or alternatively the re-establishment of vegetation may have increased the production of 'new' DOM via root leachate and fresh litter decomposition. Particulate losses from peatland systems decreased following stabilisation

of the peat surface through revegetation irrespective of gully blocking activities (Pilkington et al., 2015), as overland flow velocities are lower on vegetated peat than bare peat (Holden et al., 2008). However, the same study (Pilkington et al., 2015), and a more recent assessment of the effects of revegetation on DOC concentrations (Alderson et al., 2019), found no changes in DOC concentrations following revegetation at the headwater catchment scale.

Radiocarbon ($^{14}$C) measurements of DOC in UK upland waters indicate that the principal source of DOM in waters draining relatively undisturbed soils is recent primary production, probably formed within the last few years (Evans et al., 2014). It follows, therefore, that plant productivity and plant tissue composition and degradability, which depend both on ambient environmental conditions and species composition, may be important factors, both for DOC concentrations and the treatability of the DOM produced. Ritson et al. (2016)

showed in a laboratory-based extraction experiment that DOM leached from Sphagnum was more easily removed by a conventional coagulation process and decomposed more rapidly than DOC leached from *Molinia caerulea* or *Calluna vulgaris* litter. In addition, *M. caerulea* and *C. vulgaris* litter released more DOC per unit dry weight compared to *Sphagnum* litter. At the field scale, Armstrong et al. (2012) found that DOC concentrations in pore waters were higher in areas of blanket bog dominated by *C. vulgaris* compared to areas dominated by sedges or

*Sphagnum* species. In contrast, Parry et al. (2015) found no correlation between dominant vegetation type (differentiated into ericoid, grasses, sedges and bare peat) and stream water DOC concentrations. This may reflect





the greater heterogeneity of peatland environments at the catchment scale in comparison to single species lab scale and mesocosm experiments.

### 2.3. Plantation forestry / deforestation

It has long been recognised that forestry activities can have detrimental impacts on reservoir water quality and treatability. For example in 1984 it was shown that drainage and deforestation resulted in large sedimentation issues at Crai reservoir in south Wales (Stretton, 1984 cited in: Hudson et al. 1997). Large pulses of nutrients (N and P) can also occur after forest-felling (Neal, 2002).

To reduce the impacts of forest operations on sediment and nutrient loss and consequent raw water quality, the Forest and Water Guidelines now state that no more than 20% of a drinking water catchment should be felled in any 3 year period (Forestry Commission, 2017). In addition to this, although primarily to conserve soil carbon stocks rather than for improved water quality, the 2000 Forestry Commission guidance note on forest and peatland habitats (Patterson and Anderson, 2000) states that approval will no longer be given for forestry planting or regeneration on active raised bog or inactive raised bogs that could be restored to active bog, and areas of active
blanket bog greater than 25 ha area and > 45 – 50 cm depth.

A recent review for Yorkshire Water (Chapman et al., 2017) noted that conventional conifer site preparation on peat, peaty gley and peaty podzol soils would be expected to increase DOM concentrations. This would be largely due the implemented drainage reducing the height of the water table and the consequent increase in DOM production caused by increased aeration of surface peats (Clark et al., 2009). Jandl et al. (2007), in their review
of studies of the effect of forest management on soil carbon sequestration, highlighted two Finnish studies where DOC concentrations increased following drainage ditch installation but returned to pre-drainage levels later in the forest cycle, while Schelker et al. (2012) observed increased colour in sites being prepared for forestry in northern Sweden. Furthermore, Rask et al. (1998) reported an increase in colour in streams draining peat dominated catchments following afforestation in Finland. At a regional to national scale in the UK recent work suggests that
the presence of plantation forestry on peat soils increases DOC concentrations in streams and rivers compared to peat soils with semi-natural vegetation (Pickard et al., in prep, Williamson et al., submitted).

There has been little research on the effects of forest presence on DOM treatability, although Gough et al. (2012) evaluated DOC concentrations and $SUVA_{254}$ values in waters draining catchments forested with different tree species. They showed that leachates from pine and larch plantation yielded particularly high DOC concentrations
relative to a blanket bog control (19 and 13 mg l[-1], respectively, compared to 9 mg l[-1]). Water draining these sites also had lower $SUVA_{254}$ values (1.2 and 2.4 compared to 3.3 L mg[-1] m[-1]). This would suggest that DOM leaching from plantations dominated by these tree types may be less easily treatable than DOM from blanket bogs; a finding also seen in boreal streams where DOM in streams draining wetlands was more aromatic in character than in streams draining forests (Ågren et al., 2008). In contrast, however, they also found that leachates from spruce
plantations were lower in DOC concentration (7.3 mg l[-1]), and that SUVA values were higher (7 L mg[-1] m[-1])

A practical recommendation to minimise the impacts of site preparation on DOC concentrations in waters draining forestry sites is to retain buffer strips alongside water courses (Nisbet, 2001). An experimental study in Finland showed that a strip 10 – 50 m wide was highly effective at protecting water quality in areas where peat soils are drained for commercial forestry (Holopainen and Huttunen, 1998). However, this approach may be less effective
in relatively high-rainfall areas where much of the UK's forests are planted, and on overland-flow dominated UK blanket bogs in comparison with subsurface-flow dominated Finnish peatlands.

Evidence for effects of plantation clearfell activities on DOM concentrations is mixed. For example, in Ireland, felling increased DOC concentrations by up to 50 mg l[-1] in ditches and by a smaller but statistically significant amount in the stream draining the site (Cummins and Farrell, 2003). Effects appeared to be largely associated
with the summer peak in the DOC cycle. Significant impacts were also reported by Zheng et al. (2018) who found that DOC concentrations were elevated in a Scottish headwater catchment that drained felled areas, compared to that stream that drained moorland developed for a wind farm and where felling did not take place.

In contrast, Palviainen et al. (2014) did not observe an increase in DOM concentrations in three clear-felled catchments in eastern Finland, and various studies of the Plynlimon catchment by Neal and colleagues (e.g. Neal
et al., 2011), provide evidence of, at most, very small felling effects. Likewise, Meyer and Tate (1983) reported that DOC concentrations in a clear-felled catchment in North Carolina were lower than in a reference site and this



was attributed to reductions in organic carbon supplied by rainfall running off stems and leaves, and fresh litter (soil type not disclosed).

One aspect of forest management impacts on DOC concentrations that has received recent attention concerns the impact of plantation forest to bog restoration. Although still limited in extent within the UK, this type of restoration has been carried out for 18 years in the Flow Country in northern Scotland, and national policies on peat restoration may lead to more forest-to-bog restoration in future. Recent work in this region found that the initial impacts of forest-to-bog restoration were similar to those sometimes reported during commercial clear-felling, with increases in pore water DOC concentrations (Gaffney, 2017). Riverine DOM did not increase significantly and this was

attributed to the fact that the trees were felled to waste (the practice of leaving felled trees *in-situ* to rot) and there was less ground disturbance at the site compared with the use of machinery to extract felled timber. However, the practice of felling trees to waste has been suggested by Muller et al. (2015) to provide a potential additional DOM source as the trees slowly decompose. Gaffney (2017) showed that increased DOC concentrations persisted for at least 11 years after forest-to-bog restoration. As bog vegetation regenerated, DOC concentrations returned towards

those seen in forest control areas, although complete recovery to pre-intervention levels had still not occurred at 17 years post restoration. Extrapolation of the recovery trajectory of the Gaffney et al. (2018) data suggests that if the current trend continues DOC concentrations will not reach bog control levels until 30 years post restoration (Figure 2). Other studies have reported shorter-term increases (~4-5 years), including an assessment of forest to bog restoration of a Scottish lowland raised bog area, Flanders Moss, where baseline DOC levels were reached

within two years at one site (Shah, 2018). In a Finnish study of the impacts of forest to mire restoration, a short-term DOC concentration peak following initial restoration activity was followed by a return to reference concentrations within six years (Menberu et al., 2017).

### 2.4. Managed burning

Managed burning of peatland vegetation (Figure 3) (primarily burning heather for grouse moor management) is a

contentious issue within peatland conservation and management (e.g. Davies et al., 2016) and has been extensively reviewed over the past decade, particularly in relation to the impacts on DOM (e.g. Brown et al., 2015;Holden et al., 2012;Worrall et al., 2010), most recently by Harper et al. (2018). There is little evidence to suggest that DOC concentrations or colour increase within pore water at the plot scale following managed burns. A recent study showed no change in DOC concentrations following low and high intensity burning (Grau-Andres et al., 2019),

and in other cases plot scale DOC concentrations decreased (Clay et al., 2009;Worrall et al., 2007a). At the catchment scale it has been suggested that managed burning contributed to increases in water colour and DOC concentrations (Clutterbuck and Yallop, 2010;Yallop et al., 2010;Ramchunder et al., 2013), but these correlative studies were confounded by covariance between the extent of burning and peatland extent in the study catchments, and did not effectively take account of the concurrent effects of decreasing acid deposition (Chapman et al., 2008).

While this does not rule out the possibility of an effect of managed burning on DOC export, Holden et al. (2012) concluded that this effect was difficult to separate from the effects of land cover and vegetation, and that the apparent effects of managed burning on DOC loss might actually be an indirect consequence, resulting from changes in vegetation towards greater *Calluna* dominance. It is also worth noting that Evans et al. (2017b) found that a wildfire in Northern Ireland resulted in a temporary reduction of DOC concentrations in a downstream

monitoring lake, due to re-acidification of catchment soils following the fire. Further research into the impact of peatland wildfire on water quality is currently being undertaken in the Flow Country, northern Scotland following fires in the area in May 2019, this work is likely to be of relevance to future catchment management should future, more intense droughts increase wildfire risk.

### 3. In-lake cycling of dissolved organic matter

Lakes play an important role in fluvial carbon cycling. Up to 85% of organic carbon entering lakes and reservoirs can be lost (Algesten et al., 2004;Evans et al., 2017a), although the highest rates of DOM removal are associated with lake with very long residence times (decades to centuries) and are thus of limited relevance to water supply systems. Consequently, outflows often show reduced DOC concentrations relative to inflows (Mattsson et al.,

2005). There are a number of natural DOM loss pathways including respiration (Koehler et al., 2012;Stets et al., 2010), sedimentation (Einola et al., 2011;von Wachenfeldt and Tranvik, 2008), photo-oxidisation (via UV radiation) (Moody et al., 2013;Koehler et al., 2014) and flocculation with naturally occurring aluminium and iron (McKnight et al., 1992;Koehler et al., 2014). Hydrophobic DOM is predominantly biologically refractory but





photochemically reactive, while hydrophilic DOM is photochemically resistant and biologically labile (Berggren and del Giorgio, 2015;Berggren et al., 2018), meaning that differing loss pathways will differentially affect DOM treatability.

DOM is also generated within lakes and reservoirs via photosynthesis (production of algal exudates and release via cell lysis) and through processing of particulate matter (Tranvik et al., 2009) so that DOM concentrations at the point of abstraction from reservoirs represent the sum of these removal and generation processes. Importantly,
in-reservoir algal production, and hence within-reservoir generation of DOM is often limited by the availability of either phosphorus, nitrogen or both, so that waterbodies receiving high levels of inorganic nutrient inputs, either externally from their catchments or internally from sediments, are likely to generate additional DOM within the water column (Feuchtmayr et al., 2019;Evans et al., 2017a). DOM produced via these processes is relatively transparent and hydrophilic in comparison with DOM generated by organic rich soils, and thus presents different
challenges for treatment, particularly as the hydrophilic DOM is not easily removed through coagulation (Matilainen et al., 2010) and may lead to the need for additional capital investment in order to effectively reduce residual DOM in drinking water. Consequently, reservoirs within agricultural settings receiving high N and P loadings from fertilisers and animal wastes are particularly prone to internal production of DOM and conventional coagulation is considerably less effective than in more oligotrophic, organic-soil dominated systems.

The loss of DOM in less productive lakes often increases with lake water retention time (Evans et al., 2017a;Kohler et al., 2013). For example one of the reservoirs included in the analysis of Evans et al. (2017a) (a natural lake with a dammed outflow) was found to remove around half of all inflowing DOM. Evans et al. (2017a) showed that when sites act as DOC sinks the initial DOC loss is very rapid, with up to 40% of the inflowing DOC concentration being lost within a year. This is in agreement with previous studies showing very high initial
reactivity of DOC entering the fluvial environment (Moody et al., 2013;Köhler et al., 2002). In the case of relatively nutrient rich lakes with enhanced algal production, longer residence times tend to be associated with higher ratios of outflowing to inflowing DOC concentrations (Evans et al., 2017a). Lake retention time and the presence of upstream lakes also appears to influence long-term trends in DOC concentrations. A lake complex in Sweden showed increasing DOM concentrations over the past 20 years in the lakes with shorter retention times,
but there is little evidence of increases in those with the longest retention times since records began (Kohler et al., 2013). It may be that the availability of DOM in the downstream lakes of such complexes is controlled by DOM processing in the upstream lakes, and that increases in catchment DOC export are sequentially buffered by aquatic processes as transit times increase. Kohler et al. (2013) also demonstrated that the ratio of DOC to dissolved organic nitrogen (DON) is lower in lakes with longer residence times, which is consistent with an increasing
proportion of the DOM being derived from in-lake production This also suggests that water abstracted from reservoirs with long retention times may be more likely to form nitrogenous DBPs, and would thus represent an emerging issue to the water industry.

In their assessment of DOM in lake inflows and outflows, including those of several reservoirs, Evans et al. (2017a) concluded that any measures that can reduce N and P export from the catchment or release from
sediments, or which can strip nutrients from the water column, could provide effective mitigation for high DOM concentrations by reducing algal DOM production. For example, measures for reducing nutrient loading to lakes from the catchment (Spears and May, 2015) and bed sediments (Spears et al., 2016) can be effective in reducing algal biomass in UK lakes; although the effects on algal DOM production in relation to drinking water treatment require further assessment.

To date, this option has rarely been considered in relation to DOM-related treatment issues, although nutrient management is often considered in relation to other (taste and odour) related treatment issues. The available evidence therefore suggests that measures to reduce taste and odour problems could deliver co-benefits in relation to DOM levels.

Overall, these results suggest that measures which reduce in-reservoir DOM production, and/or favour in-reservoir
DOM removal, may be as – or perhaps more – effective than measures aimed at reducing DOM export from the terrestrial catchment. For lakes acting as DOC sources, management regimes that reduce nutrient (primarily N and P) inputs from catchments and/or internal loading of nutrients from sediment to the water column during periods of hypoxia, may be more effective than those focussed on reducing inflowing DOM concentrations directly. Restricting nutrient inputs is also likely to reduce DON concentrations relative to DOC, which has the
added benefit of reducing the formation potential of nitrogenous DBPs. However, Birk et al. (2020) suggest that rising DOM loading from the catchment may act to dampen algal responses to nutrients through light limitation





of primary production within some European lakes. If, by extension, this also limits in–reservoir DOM production then catchment interventions that relieve DOM load, but not nutrient load, may result in an increase in in-reservoir DOM production. Even in the case of less nutrient-rich water bodies, it appears that reducing N and P loadings
would be beneficial for water treatment as this is likely to restrict additional DOM formation.

## 4. Conclusions

Increasing DOM concentrations in reservoirs draining catchments dominated by organic-rich soils are a cause for concern for water companies, from both regulatory compliance and treatment cost perspectives. To a large extent
these increases appear to be a long-term large-scale phenomenon driven by reductions in atmospheric pollutant inputs, and thus beyond the direct control of catchment managers. It is also feasible that future climate change could also contribute to further increases in concentrations, particularly through changes in rainfall and temperature, although these effects require further investigation. It is therefore unrealistic to expect that recent trends can be entirely halted, yet alone reversed, through catchment management alone. To date, catchment
management initiatives, while providing clear overall restoration benefits for peatlands, do not appear to have produced a generalised solution to the challenge of increasing DOM, although there is some evidence that catchment interventions may provide benefits for DOM export in specific cases (see Table 2 for a summary of effects). We have also identified some areas where there is mounting evidence for the importance of certain catchment interventions. In particular short-term effects of forest felling and harvesting activities have repeatedly
shown to have detrimental effects on DOM concentrations. However, there is also evidence to suggest that such effects can be limited by good practice, including phased felling. Catchment interventions may also provide co-benefits such as reductions in sediment and particulate organic carbon loadings to reservoirs, reductions in greenhouse gas emissions and enhancement of biodiversity, which may justify the implementation of measures when all benefits are combined, even if the direct benefits for DOM alone may not.

In-reservoir processes, although investigated for algal and manganese control by a number of UK water companies, have been generally overlooked as a potential control of DOM concentrations at the point of abstraction. These processes may be of high importance in influencing both the concentration and quality of DOM within reservoirs, as shifts from allochthonous to autochthonous DOM production, particularly in waterbodies with a long residence time, is likely to result in a reduction in DOM treatability. Our review has highlighted that
these processes should be considered alongside catchment land management approaches by the water industry to maximise the potential for upstream solutions to rising DOM concentrations in source waters.

Our review of the published literature highlights a major current evidence gap of importance to the water industry: the quantification of the impacts of catchment management on DOM concentration and treatability at the point of abstraction. The size of the research challenge with respect to the necessary spatial and temporal scale and need
for robust Before-After-Control Impact (BACI) of any field experiment cannot be underestimated, and perhaps explains in part the current dearth of reliable information. This is particularly pertinent when changes in water chemistry may take a number of years to be seen, depending on catchment dynamics and within reservoir processes.

Overall, this review suggests that management of the terrestrial catchment may contribute to mitigating the
impacts of regional-scale DOM increases in some circumstances. Activities that reduce the production of terrestrial DOM at source therefore have a potential role in mitigating observed DOM increases in raw water supplies. Catchment management measures that reduce in-reservoir DOM production, or favour in-reservoir DOM removal, may be as or more effective, particularly with respect to more agriculturally influenced systems. More generally, it seems clear that catchment management should be considered part of the response strategy to rising
DOM levels, and as part of a process to improve the resilience of source waters, not a panacea. It is therefore important that the water industry also develops effective tools to predict likely future DOM levels resulting from a combination of large-scale and catchment-scale drivers, to ensure that investments in both catchment management measures and DOM treatment infrastructure are correctly targeted, integrated, timely and cost-effective.


The authors declare that they have no conflict of interests.





Acknowledgements:

We thank staff from Scottish Water, United Utilities, Yorkshire Water, Irish Water and Dŵr Cymru Welsh Water for their informative discussions and comments on early drafts of this manuscript. Discussions with Nadeem Shah and Tom Nisbet regarding work being undertaken by Forest Research are also gratefully acknowledged. This work was funded by a NERC Environmental Risks to Infrastructure Innovation Programme grant and Scottish Water.



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

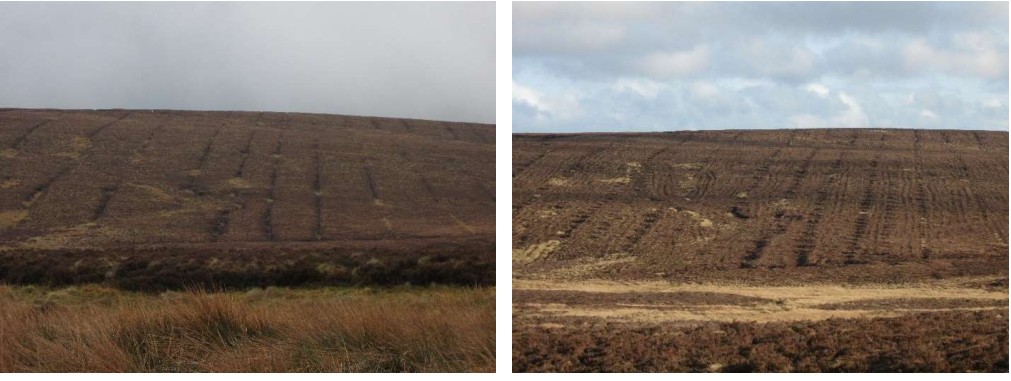

**Figure 1: Drainage ditches before (left) and after (right) blocking on a blanket bog in North Wales, the ditches run down the slope and individual dams can be seen crossing the ditches (Photos: Chris Evans).**

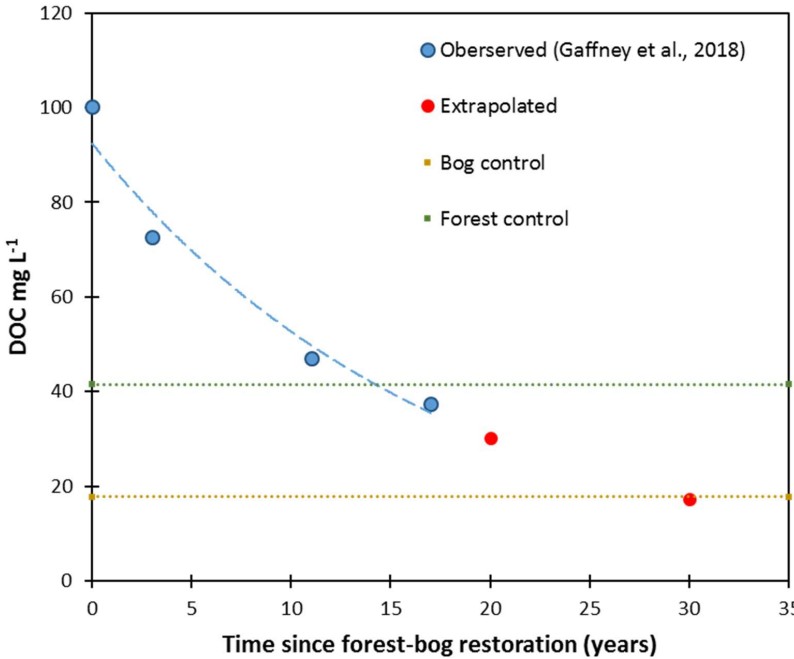

**Figure 2: Measured and projected timeline for effects of forest to bog restoration in the Flow Country northern Scotland on DOC concentrations within peat surface pore water (Measured data from Gaffney et al. (2018)). Blue data points are measured pore water DOC concentrations, red points are the projected DOC concentrations derived from fitting an exponential curve to the observed data, the green line shows mean pore water DOC within the forest plantation control site, while the brown line shows mean pore water DOC at the bog control site.**





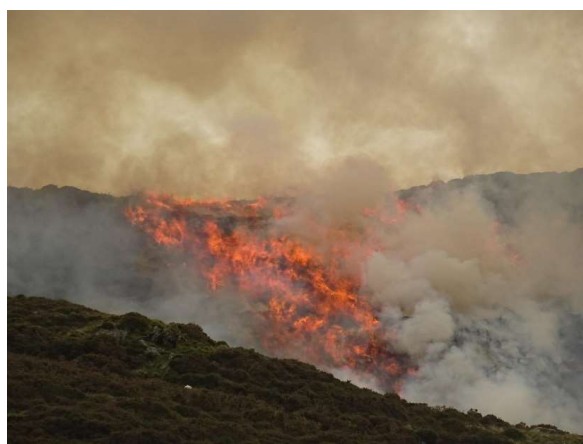

**Figure 3: Burning of vegetation on peat in North Wales (Photo: Chris Evans).**

**Table 1: a summary of the impacts of drainage ditch blocking on DOC concentrations and fluxes from peatlands, reported in increasing time since ditch blocking. BA = Before/After, CI = Control/Intervention**


| Reference | Location | Sampling scale | Concentration or flux measured | Time since ditch blocking | Experimental Design | Change since ditch blocking |
|---|---|---|---|---|---|---|
| Worrall et al. (2007b) | UK, blanket bog | Ditches | DOC concentration | 7 months | BACI | 100% increase in DOC concentration. |
| Turner et al. (2013) | UK, blanket bog | 0 and 1st order ditches | DOC concentration and flux | 1 year | BACI | DOC concentration decreased by 2.5% compared to control, DOC flux decreased by 2.2 – 9.2% as a result of decreased water export. |
| Gibson et al. (2009) | UK, blanket bog | Ditches | DOC concentration and flux | 1 year | Primary CI | DOC concentrations unchanged, Water flux decreased by 39% meaning DOC flux also declined by the same amount. |
| Wilson et al. (2011) | UK, blanket bog | Ditches and headwater streams | DOC concentration and flux | 2 years | BACI | DOC concentrations unchanged, fluxes were 88% lower in streams draining ditch-blocked catchments due to much lower estimated water export (see text for further discussion). |
| O'Brien et al. (2008) | UK, blanket bog | Headwater streams | DOC flux and water colour | 2 years | BACI | Water colour was unchanged. Fluxes decreased by 24% in streams as a result of decreasing water export. |
| Menberu et al. (2017) | Finland fen, pine mire and spruce mire | Pore water | DOC concentration | 3 years | BACI | 41% reduction in DOC concentration. |





| Evans et al. (2018) | UK, blanket bog | Ditches | DOC concentration | 4 years | BACI | No change |
|---|---|---|---|---|---|---|
| Wallage et al. (2006) | UK, blanket bog | Pore water | DOC concentration | 5 years | CI | DOC concentration lower in porewaters adjacent to blocked ditches (69% lower compared to open ditches) |
| Haapalehto et al. (2014) | Finland, raised bog | Pore water | DOC concentration | 5 years and 10 years | Chronosequence | DOC concentration approx. 10% lower in sites 5 years post restoration and 25% lower in sites 10 years post restoration |
| Haapalehto et al. (2014) | Finland, raised bog | Ditches | DOC concentration | 5 years and 10 years | Chronosequence | Concentrations approx. 75% higher in sites 5 years post restoration and 50% higher in sites 10 years post restoration |
| Armstrong et al. (2010) | UK, blanket bog | Ditches | DOC flux | 7 years | CI | No change in DOC flux |
| Strack et al. (2015) | Canada, bog | Pore water and ditch water | DOC concentration | 10 years | CI | No change in pore water DOC concentration. Ditch water concentrations were similar in spring and summer and up to 30% lower in the restored site in autumn. |
| Armstrong et al. (2010) | UK, blanket bog | Ditches from a survey in Northern England and Northern Scotland | DOC concentration | 6 months to 18 years | Survey | DOC concentrations 28% lower on average in blocked drains compared to unblocked drains. |
| Holl et al. (2009) | Germany, ex-fenland extraction site | Pore water | DOC concentration | 20 years | CI | DOC concentrations 37% lower at restored site compared to drained site. |
| Urbanova et al. (2011) | Czech Republic, bog | Pore water | DOC concentration | NA comparison between drained and intact sites | CI | No difference in concentration between intact and moderately degraded site, 50% higher DOC concentrations at highly degraded site. |

**Table 2: summary of the published impacts of catchment management activities on DOM concentrations and treatability. Numbers in brackets refer to the number of studies showing that effect in each case. Colour coding shows whether the overall conclusion is that effects are positive (green), no / limited change (yellow), or negative (red); while the colour intensity refers to confidence in the conclusions drawn – pale colours indicate lower confidence, while bold colours indicate higher confidence.**

| Catchment intervention | Impact on DOC concentration | Impact on DOM treatability |
|---|---|---|
| Ditch blocking | Increase (3)<br>No change (7)<br>Decrease (7) | Increase (3)<br>No change (5)<br>Decrease (2) |
| Revegetation to grass species | Increase (1)<br>No change (3) | Decrease (1) |



| | | |
|---|---|---|
| Revegetation to heather | Increase (1) No change (1) | Decrease (1) |
| Revegetation to *Sphagnum* | | Increase (1) |
| Forest planting | Increase (5) | Species dependent increase (1) Decrease (2) |
| Clearfell and forest to bog conversion | Increase (5) No change (2) Decrease (2) | |
| Managed burning | Increase (3) No change (1) Decrease (3) | |