# Peer review of "Will UK peatland restoration reduce dissolved organic matter concentrations in upland drinking water supplies?"

_Hydrology and Earth System Sciences, 2020_

## Referee Comment (RC1) · Eleanor Jennings (Referee) · 3 Jan 2021

General comments:

This review paper deals with the potential for reduction of dissolved organic matter (DOM) concentrations in potable water sources due to the implementation of catchment restoration measures. The authors note that such measures are now starting to be implemented by water treatment companies, but that the evidence for their success in reducing concentrations is still lacking. The paper, therefore, is of immediate relevance and of use to water managers and policy makers. It also has the potential to highlight new research areas which could address knowledge gaps on this topic. Overall, it

is generally well written and achieves its aims in terms of the review of the literature on catchment management in peatlands and effects on DOM concentrations. It is very informative and thought-provoking. However, while I consider that it is worthy of publication, there are points that do need to be addressed. I outline these below, and also note other more minor corrections for the authors.

Specific comments:

1. Section 1.1 has much relevant information but needs to be better supported by references to inform the reader, especially as the paper is based on a review of the existing literature. For example, the section in lines 48 to 60 on DOM composition is not currently supported by any literature. Similarly the sentences at lines 84-89 and lines 93-94 need supporting references.

2. Section 2 is the strongest section of the paper. It is well referenced and written, and argues its points well. It provides an excellent assessment of the available studies. Table 1 should be referred to more throughout this section to guide the reader. The information contained in this table is central to the points raised but it is referred to only once in the text. I also suggest that Table 2 be brought up into the later part of Section 2. It currently appears only in Conclusions, but logically would come after the review of the studies has been presented.

3. Table 2 also needs some amendments. I strongly suggest that the authors confine themselves to three colours and omit any indication of confidence. As they note, there are still a limited number of studies on this topic and I do not consider that they have enough evidence to infer a confidence level as indicated by the darker red colour. This table should also be expanded to include a column that showed key supporting references. These are not so great in number that they cannot be included. A numbering system for references in Table 2 could also link back to Table 1 where appropriate.

4. Overall, I consider that structure of the paper could be reordered to better lead the reader through the complexities of the issues. If the paper aims to also deal with processes in lakes, there is currently some material in Section 3 on in-lake processing of DOM which would be more useful earlier in the paper. I suggest that it be incorporated into a new short section on processing of DOM in both fluvial and lacustrine systems that comes after 1.1.

5. The focus in Section 3 on in-lake processes is not reflected in the current title and therefore the implied scope of the paper, although it is included in the aims. This was a little confusing on first reading the paper. I strongly consider that Section 3 should be restructured and retitled to focus on knowledge gaps in general. In-lake/reservoir processing of DOM in relation to water treatment should be included as one of these gaps, together with topics that need to be addressed in catchment processing of DOM, catchment management and indeed processing of DOM in river networks. One current issue with the paper is that the section on in-lake processing reviews only a small section of the vast literature on carbon cycling in lakes. The implications of in-lake processing for water treatment and in particular carbon cycling could be argued to actually merit a separate review paper. The number of papers on C cycling in lakes and reservoirs has increased hugely in the last decade, and the DOM cycling will be affected by a range of additional processes not referred currently to by the authors. Changing this section to address gaps in general in relation to management of DOM would allow this material to remain, and would give a better structure to the paper.

6. Figure 2 is one of the weaker parts of the paper. It is based on four data points extracted from another publication (Gaffney et al., 2018) and presents two 'new' data points based on an equation using those data. However, I consider that the approach is not sound. The relationship presented is based on the mean porewater DOC concentrations, abstracted from the original chronosequence study. The current authors then projected the two new values for 20 and 30 years post-restoration for that site apparently using an exponential equation based on those four summary values alone. The original paper indicated larger datasets that were summarised as boxplots, where these mean values were also indicated. Those original plots indicated a relatively wide

degree of uncertainty which is not taken into account in their use in Figure 2. There is no indication here that the original full datasets were available to the authors, nor is the equation used presented. I do not consider that the authors should take these mean values and extrapolate future trends, especially 1. without taking account of any uncertainty in the original data, and 2. given the range of processes that could influence those future trends, processes that they describe comprehensively in this paper. Even in the original paper, Gaffney et al. (2018) were only willing to state that their results 'suggest that at least >17 years is likely required for complete recovery of water chemistry to bog conditions'. Figure 2 should be removed and the text amended, but should include the point that this study showed that restoration may take at least 17 years i.e. multiple decades.

7. The paper would, however, benefit hugely from a conceptual figure that could illustrate the mechanisms that control DOM concentrations and quality, including those related to catchment restoration. This figure could then be used to guide the structure of the revised paper.

8. Line 390: the authors state that 'Overall, these results suggest that measures which reduce in-reservoir DOM production, and/or favour in-reservoir DOM removal, may be as – or perhaps more – effective than measures aimed at reducing DOM export from the terrestrial catchment.' This is a strong statement and could be supported a more concrete way, for example a table that compares the published reductions in DOM concentrations in reservoirs due to the cited measures.

9. I also consider that the authors do not currently highlight their own analysis and conclusions based on Section 2 enough in the abstract, and should rewrite with this in mind.

Technical comments:

Line 45: sentence is missing a full stop.
Line 56: suggest that dissolved organic matter should be in lower case, with abbreviation in capitals.

Line 69: the heading here needs to be moved down a line.

Line 80: I suggest that these lines follow on directly after the introduction of THMs in the paragraph before, rather than in a new paragraph.

Line 164: I question the use of 'most' when n = 4; I suggest state for example 'three of the four studies. . .'.

Line 212: this point needs a supporting reference.

Line 335: the heading, or subheading title (subheading if this section is changed as suggested in the Specific comments), used for this should indicate that it applies to lakes and to reservoirs. This section should start by making a point on how many sources of potable water are lakes/reservoirs, or what % of the population get their water from lakes/reservoirs (information on whether this true for the UK would support this).

Line 336: 'Lakes play an important role in fluvial carbon cycling'. I suggest that the term 'fluvial carbon cycling' should be changed. This carbon may be exported from a river to a reservoir but catchment soils are likely to be the more dominant source. This term could be taken to imply that the carbon originated in a river/stream.

Line 345: differing loss pathways will differentially affect DOM treatability – this is an important point.

Line 337: the authors use the term 'can be lost'. Please be more specific in this sentence on the processes that you are referring to. You do go on to give more detail further on, but I would expand this sentence here for clarity.

Line 355: change to 'this hydrophilic DOM for clarity'.

Line 382: suggest this sentence come in the previous paragraph as it continues that

point.

Line 395: the point supported here by Birk et al. 2020 has long been recognised. I suggest that you refer to some of the other literature on this effect.

Line 402: suggest this should read 'increases in concentrations and changes in the quality of DOM. . .'.

Line 420: here they refer to 'algal and manganese control by a number of UK water companies'. I wonder why this point is here in Conclusions and is this not included with detail in the in-lake processes section?

Line 807-809, Figure 1 legend: clarify where on image the dams are visible for 'individual dams can be seen crossing the ditches in image on the left'. Some arrows could be used here to indicate a dam.

Line 796: The reference 'Worrall, F., Clay, G. D., Marrs, R., and Reed, M.: Impacts of burning management on peatlands, 2010' is lacking some of required detail.

Table 1. I suggest that BA be separated by a space or comma from CI where used together in the column 'Experimental design'. Also the term 'Primary CI' is not clear to me. For the line O'Brien et al., clarify that you refer to 'DOC fluxes' in the last column. For the line Urbanova et al. add a comma or semi-colon after NA (note that NA should also be defined in the legend).

---

## Editor Comment (EC1) · Matthew Hipsey (Editor) · 7 Apr 2021

Whilst we await a further reviews, I request the authors to provide a response to the questions/comments so as to provide an opportunity for further discussion.

---

## Referee Comment (RC2) · Anonymous Referee #2 · 3 May 2021

Overall, the paper covers an interesting topic that is within the scope for HESS. It is interesting to acknowledge that DOC has been increasing in the UK and is impacting water treatment. The real substance of the paper is assessing literature about catchment management benefits on DOC export, and whether this can be an effective strategy to reduce DOC in waters.

Whilst the paper is relevant and covering and interesting topic, I do not believe it is yet ready for publication. The structure of the paper has problems in my view that detract from the story and I found many sentences and paragraphs lacking the necessary level of precision and accuracy to really support the claims. There are also typographical

and editorial issues, making it seem it wasn't thoroughly proof-read before submission.

The paper is reviewing literature, mainly specific to material UK, but needs to more obviously try to bring together the story and synthesize the results. After reading the conclusions I was not convinced about the scale or nature of the benefits from different management actions, and yet strong statements are made recommending catchment management.

I also found the section on in-lake processing to be too generic without a clear focus – the message is that lake processes can mediate (increase or decrease) incoming DOC before it hits the treatment plant, but I find lacks any clear conclusion, other than making a connection with nutrient management. What is the typical difference between input DOC and offtake DOC? Even as a broad range to give some indication would make the review much more powerful.

For a review like this I refer to the authors to a document like this one: https://www.sciencedirect.com/science/article/pii/S221501611930353X "Method for conducting systematic literature review and meta-analysis for environmental science research" by Mengist et al.

Ideally adding some conceptual models or diagrams bringing together the ideas in the paper would be very beneficial and help add meaning to the literature, which is currently quite mixed in terms of its results, as a way explaining some of the differing reports.

For this reason, I think the paper needs some major revision and re-working to give it improved focus and flow, refine some of the text, and to highlight key take home findings that are supported by the data. It could certainly be a strong paper with some further development, and I have provided some more specific comments below to support the recommendation.

Abstract

The two opening sentences could be better re-written to highlight the problem to the water industry. Currently it is asserted to be a problem and implied to be associated with colour, but I suspect the concern is related to treatment by-products.

For the sentence "One of the primary evidence gaps is the extent to which catchment management is capable of influencing DOM concentrations at the point of abstraction, field studies rarely extending beyond sub-catchment or stream scale." . . .needs rewording – the first part makes sense but the second half is a fragment.

Given the lack of evidence is discussed to establish the link between management and response, including something like "research priorities were therefore established" would be logical, rather than ending on the result that evidence is insufficient. Further, the last sentence and second-to-last sentence seem to contradict each other. One says insufficient for wide spread application and then it says the measures have good potential. I think these two sentences could benefit from some rewording to avoid confusion, and make the outcome of the paper more clear.

Introduction

The opening sentence seeks to make the claim DOC is increasing from 1980- to present, but has a citation that is 1989 and 2001. The rest of the paragraph could be polished in terms of wording. Second sentence repeats the claim of 1980's beginning. Last sentence suggests a more recent reference which is good, but comes after a sentence related to international evidence. Overall, I the paragraph is somewhat awkward, and could more clearly make the argument that a) UK rivers have experienced rising DOC from 1980 to present, and b) similar trends have also been seen elsewhere in the world. The paragraph also doesn't give much quantitative evidence of how much things have changed. It is probably beneficial to include here a figure (or reproduce a figure?) allowing readers to see an example of what this increasing trend looks like.

Lines 48-60 are describing DOM but no references support the statements.

Section 1.2 heading issue

Line 73 – why is E coli mentioned? Is it related to DOM? If not provide context. Does this need

Paragraph 1 would sit better after the DBP/THM paragraph, before the paragraph starting as "Higher concentrations . . ." Line 91 – Opening line would benefit from a reference

Section 1.3 is interesting. I think it could be refined to more clearly point out that there are drivers associated with (geo)chemical changes and those associated with hydrological changes; currently they are slightly intertwined. The section could end by summarising the research unknowns that remain. The climate change sentence at the end seems to be less relevant to this section since it is explaining the past and I don't think the single sentence does this issue justice.

Line 140 – I think a sentence that is pivoting like this needs to being with Also, or Further or In addition.

Line 142 – It is asserted here that catchment management activities are not seen at the point of abstraction (presumably you mean at the reservoir outlet?) but it is not obvious from the prior text this is established. Is there a published paper saying this, or just a "hunch"? Section 2 obviously goes into this, but in this case the text is out of order.

Line 147 – ditch blocking mentioned here but not above in the catchment management section, so seems out of context.

I find the aims statement buried in sub-section 1.4, quite deep into the manuscript, to be somewhat awkward. The aims statement is weak in that the aim of the paper "bring together information" and "contribute to our understanding" and "go on to examine". These aims lack specificity and are overly general in my view, making it difficult for the reader to clearly understand what the outcomes of the paper will be. Whilst I acknowledge it is a review paper, a good review can still have specific aims. E.g "Is

there evidence that . . .". or "The review is used to develop a conceptual model. . .."

Line 155 – this paragraph flows well

Line 166 – the line "While these results are persuasive, they do not necessarily imply that effects will be translated through to surface waters and ultimately to the point of abstraction" seems unnecessary at this point, between describing pore water changes and ditch water changes.

Line 175 – I struggled to follow this logic. If a study was from a hydrological point of view then there is a view that DOC decreases? But Wilson says that DOC load went down but not conc? This section could benefit from the authors make a conceptual diagram to synthesise the results.

Line 207 – I don't think the last two sentences of this paragraph are relevant for an international journal. Line 331 – This sentence is not really adding anything – It is great people are doing more work, but in this paper it would be just better to present published findings.

Section 3 – this section reads reasonably well. The key is whether the creation or consumption of DOM is big or small relative to the a) the observed increases mentioned in the introduction, and b) the catchment management activities. I cant tell from reading this.

Conclusions – A lot of the conclusions seems more like opinion, and I'm looking for more specific summary here – scientifically what is the evidence, per km2 of land, that DOC will go up or down for a given intervention? Is one intervention more effective? How does land management actions compare to in-lake processes in potential amounts of DOC removal?

Finally, is there a role for models to help compute a DOM budget? It seems that modelling is overlooked, but can be useful for assessing this issue and so should be in the review.

---

## Author Comment (AC1) · 30 Jun 2021

General comments: This review paper deals with the potential for reduction of dissolved organic matter (DOM) concentrations in potable water sources due to the implementation of catchment restoration measures. The authors note that such measures are now starting to be implemented by water treatment companies, but that the evidence for their success in reducing concentrations is still lacking. The paper, therefore, is of immediate relevance and of use to water managers and policy makers. It also has the potential to highlight new research areas which could address knowledge gaps on this topic. Overall, it is generally well written and achieves its aims in terms of the review

of the literature on catchment management in peatlands and effects on DOM concentrations. It is very informative and thought-provoking. However, while I consider that it is worthy of publication, there are points that do need to be addressed. I outline these below, and also note other more minor corrections for the authors. We thank the reviewer for their positive and helpful comments on the manuscript. We have responded to specific comments in-line below

Specific comments: 1.1. Section 1.1 has much relevant information but needs to be better supported by references to inform the reader, especially as the paper is based on a review of the existing literature. For example, the section in lines 48 to 60 on DOM composition is not currently supported by any literature. Similarly the sentences at lines 84-89 and lines 93-94 need supporting references. We agree that this section would benefit from more references to the literature and suggest including the following: i) For DOM composition (lines48-60): Anderson et al 2019, Thurman 1985, Tipping et al 2009 ii) For chloramination (line 84): Norman et al 1980 iii) Regarding customer perception of coloured water (line 89) Ritson et al 2014 iv) Chow et al 2005 provides an example of a water industry specific reference regarding filter size and definition of DOC, which we think would be a useful addition to this paragraph. v) For DOM and bacterial regrowth in water pipes (line 87): Prest et al 2016 vi) The costing information (lines 93-94) has resulted from numerous discussions with water industry representatives as part of the NERC funded Freedom project that funded this review. The outcomes of these discussions are currently in the final editing process before publication as a series of briefing notes aimed at the UK water industry so we would suggest referencing Pickard et al 2021 (full reference: Pickard, A.E., Chapman, P.J., Williamson, J., Spears, B.M., Banks, J., Bullen, C., Leith, F., Gaston, L., Moody, C., and Monteith, D.: Rising concentrations of dissolved organic matter in drinking water supplies: can peatland restoration help? FREEDOM-BCCR briefing note I to the water industry. UKRI SPF UK Climate Resilience programme – Project no. NE/S016937/2. 2021.)

1.2. Section 2 is the strongest section of the paper. It is well referenced and written,

and argues its points well. It provides an excellent assessment of the available studies. Table 1 should be referred to more throughout this section to guide the reader. The information contained in this table is central to the points raised but it is referred to only once in the text. I also suggest that Table 2 be brought up into the later part of Section 2. It currently appears only in Conclusions, but logically would come after the review of the studies has been presented. We will move the location of Table 2 and refer to it within the relevant sub-sections of Section 2. We agree that logically it would be better placed earlier in the paper.

1.3. Table 2 also needs some amendments. I strongly suggest that the authors confine themselves to three colours and omit any indication of confidence. As they note, there are still a limited number of studies on this topic and I do not consider that they have enough evidence to infer a confidence level as indicated by the darker red colour. This table should also be expanded to include a column that showed key supporting references. These are not so great in number that they cannot be included. A numbering system for references in Table 2 could also link back to Table 1 where appropriate. This is a good idea, and we will modify Table 2 in line with the reviewer's suggestions

1.4. Overall, I consider that structure of the paper could be reordered to better lead the reader through the complexities of the issues. If the paper aims to also deal with processes in lakes, there is currently some material in Section 3 on in-lake processing of DOM which would be more useful earlier in the paper. I suggest that it be incorporated into a new short section on processing of DOM in both fluvial and lacustrine systems that comes after 1.1. We will add a section on DOM processing in rivers and lakes into the introduction.

1.5. The focus in Section 3 on in-lake processes is not reflected in the current title and therefore the implied scope of the paper, although it is included in the aims. This was a little confusing on first reading the paper. I strongly consider that Section 3 should be restructured and retitled to focus on knowledge gaps in general. In-lake/reservoir processing of DOM in relation to water treatment should be included as one of these

gaps, together with topics that need to be addressed in catchment processing of DOM, catchment management and indeed processing of DOM in river networks. One current issue with the paper is that the section on in-lake processing reviews only a small section of the vast literature on carbon cycling in lakes. The implications of in-lake processing for water treatment and in particular carbon cycling could be argued to actually merit a separate review paper. The number of papers on C cycling in lakes and reservoirs has increased hugely in the last decade, and the DOM cycling will be affected by a range of additional processes not referred currently to by the authors. Changing this section to address gaps in general in relation to management of DOM would allow this material to remain, and would give a better structure to the paper. We agree with the reviewer that a complete review of carbon cycling in lakes is beyond the scope of this paper and would make an interesting paper of their own. We will change the structure of this section to cover evidence gaps specific to the water industry as suggested (and move some of the introductory information on in-lake processes to the introduction as outlined in point 1.4 above). This will improve the flow of the paper, and along with a conceptual diagram (see point 1.7 below) will make the paper easier to follow.

1.6. Figure 2 is one of the weaker parts of the paper. It is based on four data points extracted from another publication (Gaffney et al., 2018) and presents two 'new' data points based on an equation using those data. However, I consider that the approach is not sound. The relationship presented is based on the mean porewater DOC con-centrations, abstracted from the original chronosequence study. The current authors then projected the two new values for 20 and 30 years post-restoration for that site apparently using an exponential equation based on those four summary values alone. The original paper indicated larger datasets that were summarised as boxplots, where these mean values were also indicated. Those original plots indicated a relatively wide degree of uncertainty which is not taken into account in their use in Figure 2. There is no indication here that the original full datasets were available to the authors, nor is the equation used presented. I do not consider that the authors should take these
mean values and extrapolate future trends, especially 1. without taking account of any uncertainty in the original data, and 2. given the range of processes that could influence those future trends, processes that they describe comprehensively in this paper. Even in the original paper, Gaffney et al. (2018) were only willing to state that their results 'suggest that at least >17 years is likely required for complete recovery of water chemistry to bog conditions'. Figure 2 should be removed and the text amended, but should include the point that this study showed that restoration may take at least 17 years i.e. multiple decades. On reflection, and although authors had access to the data contained within the Gaffney et al paper, we agree with the reviewer and will remove Fig 2 as it is based on limited data, and from one location, which may not be representative of other areas. We will also add reference to a new paper on this topic (Howson et al 2021) and compare their results to those of Gaffney in the text only.

1.7. The paper would, however, benefit hugely from a conceptual figure that could illustrate the mechanisms that control DOM concentrations and quality, including those related to catchment restoration. This figure could then be used to guide the structure of the revised paper. We have attached a potential conceptual diagram to this response. This will be further neatened provided it is felt that we have covered the main points. We would put this diagram in the introduction and add the aims up front to aid the reader in following the text through. This would be included with a paragraph as follows before section 1.1: The past decade has seen peatland restoration become an integral part of the UK's environment policies, including the development of an English Peat Strategy, the Welsh Government's commitment to the restoration of all semi-natural peatlands and Scotland's national peatland plan, because of the potential role of restoration in improving biodiversity, carbon storage and natural water management. As nearly three quarters of the storage capacity of drinking water reservoirs in the UK is derived from water draining peaty areas (Xu et al 2018) the UK water industry has investigated whether peatland restoration and management could improve water quality at source in order to reduce treatment costs and risk of regulatory failure (See schematic). This review summarises the available evidence regarding the effects

of peatland catchment management on water quality, highlights the current evidence gaps and suggests priorities for future investigation. Then remove lines 152-159.

1.8. Line 390: the authors state that 'Overall, these results suggest that measures which reduce in-reservoir DOM production, and/or favour in-reservoir DOM removal, may be as – or perhaps more – effective than measures aimed at reducing DOM export from the terrestrial catchment.' This is a strong statement and could be supported a more concrete way, for example a table that compares the published reductions in DOM concentrations in reservoirs due to the cited measures. This comment related to the magnitude of change seen in some sites reviewed in Evans et al 2017. We feel that this could be worded better and state that there is currently a knowledge gap with regards to the impact of catchment management on in-reservoir carbon cycling, and the extent to which measures which reduce in-reservoir DOM production and/or favour in-reservoir removal of DOM. This knowledge gap could be explored further by the UK water industry, particularly in areas where previous catchment management has not shown an improvement in water quality.

1.9. I also consider that the authors do not currently highlight their own analysis and conclusions based on Section 2 enough in the abstract, and should rewrite with this in mind. See comment to Reviewer 2, in section 2.1

Technical comments: Line 45: sentence is missing a full stop. Agreed Line 56: suggest that dissolved organic matter should be in lower case, with abbreviation in capitals. Agreed Line 69: the heading here needs to be moved down a line. Agreed Line 80: I suggest that these lines follow on directly after the introduction of THMs in the paragraph before, rather than in a new paragraph. Agreed, change as suggested. Line 164: I question the use of 'most' when n = 4; I suggest state for example 'three of the four studies. . .'. Change as suggested

Line 212: this point needs a supporting reference. We will include Stimson et al (2017), and refer the reader to further references within this paper. We will also include further
reference to this paper in the section on revegetation.

Line 335: the heading, or subheading title (subheading if this section is changed as suggested in the Specific comments), used for this should indicate that it applies to lakes and to reservoirs. This section should start by making a point on how many sources of potable water are lakes/reservoirs, or what % of the population get their water from lakes/reservoirs (information on whether this true for the UK would support this). Xu et al 2018 estimate that 72.5% of the storage capacity of UK reservoirs, or 1.56 billion cubic metres of drinking water per year, derive from organic soils in the UK, which supports 43% of the UK population. There are over 450 large dams for public water supply in the UK of which 80% are in upland areas (CIWEM 2011). In the UK surface water supplies 70% of drinking water (water.org.uk) Rather than bringing this information in here, I think that this information, alongside the suggested schematic, would be better as an opening paragraph showing the scale of the problem and why it is worth looking into. I think this would help with the comments that the structure needs improving and would assist the reader in following the article.

Line 336: 'Lakes play an important role in fluvial carbon cycling'. I suggest that the term 'fluvial carbon cycling' should be changed. This carbon may be exported from a river to a reservoir but catchment soils are likely to be the more dominant source. This term could be taken to imply that the carbon originated in a river/stream. We agree that the catchment soils are likely to be the more dominant source, especially in the upland landscapes we are referring to in this review. We would therefore suggest "catchment carbon cycling" as a more inclusive term.

Line 345: differing loss pathways will differentially affect DOM treatability – this is an important point. Thank you. We will make sure this point is highlighted in the conclusions.

Line 337: the authors use the term 'can be lost'. Please be more specific in this sentence on the processes that you are referring to. You do go on to give more detail

further on, but I would expand this sentence here for clarity. We suggest changing the sentence to: up to 85% of OC entering lakes & reservoirs can be recycled into biomass, held in sediments or emitted to the atmosphere, retaining the original references.

Line 355: change to 'this hydrophilic DOM for clarity'. Agreed

Line 382: suggest this sentence come in the previous paragraph as it continues that point. On reflection, yes it does belong with the previous paragraph.

Line 395: the point supported here by Birk et al. 2020 has long been recognised. I suggest that you refer to some of the other literature on this effect. We will add further example references including Bracchini et al 2006, Carpenter 1998, and Karlsson et al 2009.

Line 402: suggest this should read 'increases in concentrations and changes in the quality of DOM. . .'. Agreed

Line 420: here they refer to 'algal and manganese control by a number of UK water companies'. I wonder why this point is here in Conclusions and is this not included with detail in the in-lake processes section? This point was included as water companies have looked into whether they can control in-lake processes, or use differing draw off depths, but to our knowledge these have been investigated for control of manganese and algae in the raw water entering the WTW rather than whether they make a difference to DOM concentrations. The part of the sentence referring to algae and Mn could be removed without changing the meaning.

Line 807-809, Figure 1 legend: clarify where on image the dams are visible for 'individual dams can be seen crossing the ditches in image on the left'. Some arrows could be used here to indicate a dam. Agreed, we will add these.

Line 796: The reference 'Worrall, F., Clay, G. D., Marrs, R., and Reed, M.: Impacts of burning management on peatlands, 2010' is lacking some of required detail. The reference should read Worrall, F., Clay, GD., Marrs, R. and Reed, M. (2010) Impacts

of burning management on peatlands. IUCN Peatland Programme Commission of Enquiry on Peatlands. We will amend.

Table 1. I suggest that BA be separated by a space or comma from CI where used together in the column 'Experimental design'. Also the term 'Primary CI' is not clear to me. For the line O'Brien et al., clarify that you refer to 'DOC fluxes' in the last column. For the line Urbanova et al. add a comma or semi-colon after NA (note that NA should also be defined in the legend). We will change to BA CI rather than BACI. Primary should read primarily – as in I think they did do one measurement prior to ditch blocking but not an extended period of baseline measurements. It would probably be clearer to refer to this as CI only.
* * *
[Figure]

**Anthropogenic pressures**

- Erosion from degraded peat
- DOC production from exposed degraded peats
- Release of DOC from forest litter
- Differential DOC release from vegetation types

Agricultural release of nitrogen and phosphorus into water stimulates algal growth and production of "hard to treat" DOC

**Regulatory pressures**
Colour < 20 hazen
THMs < 100 µg l⁻¹

Abstraction

Distribution

**Catchment management**

reduce DOC export:

- Peat restoration
- Fire control
- Forest management
- Vegetation management

reduce nutrient loading:

- Reduced fertiliser use
- Nutrient management
- Riparian buffers
- Vegetation management

**Reservoir management :**

reduce DOC production:

- P stripping
- Reduced internal P loading
- Coagulation

**Water treatment**

Remove DOC from raw water:

- Coagulation
- Ozonation
- UV filtration

**Fig. 1.**

---

## Author Comment (AC2) · 30 Jun 2021

Anonymous Referee #2 Overall, the paper covers an interesting topic that is within the scope for HESS. It is interesting to acknowledge that DOC has been increasing in the UK and is impacting water treatment. The real substance of the paper is assessing literature about catchment management benefits on DOC export, and whether this can be an effective strategy to reduce DOC in waters. Whilst the paper is relevant and covering and interesting topic, I do not believe it is yet ready for publication. The structure of the paper has problems in my view that detract from the story and I found many sentences and paragraphs lacking the necessary level of precision and accuracy to really support the claims. There are also typographical and editorial issues, making it seem it wasn't thoroughly proof-read before submission. The paper is reviewing literature, mainly specific to material UK, but needs to more obviously try to bring together the story and synthesize the results. After reading the conclusions I was not convinced about the scale or nature of the benefits from different management actions, and yet strong statements are made recommending catchment management. I also found the section on in-lake processing to be too generic without a clear focus – the message is that lake processes can mediate (increase or decrease) incoming DOC before it hits the treatment plant, but I find lacks any clear conclusion, other than making a connection with nutrient management. What is the typical difference between input DOC and offtake DOC? Even as a broad range to give some indication would make the review much more powerful. For a review like this I refer to the authors to a document like this one: https://www.sciencedirect.com/science/article/pii/S221501611930353X "Method for conducting systematic literature review and meta-analysis for environmental science research" by Mengist et al. Ideally adding some conceptual models or diagrams bringing together the ideas in the paper would be very beneficial and help add meaning to the literature, which is currently quite mixed in terms of its results, as a way explaining some of the differing reports. For this reason, I think the paper needs some major revision and re-working to give it improved focus and flow, refine some of the text, and to highlight key take home findings that are supported by the data. It could certainly be a strong paper with some further development, and I have provided some more specific comments below to support the recommendation. We thank the reviewer for their time and helpful comments on the manuscript.

2.1: Abstract The two opening sentences could be better re-written to highlight the problem to the water industry. Currently it is asserted to be a problem and implied to be associated with colour, but I suspect the concern is related to treatment by-products. For the sentence "One of the primary evidence gaps is the extent to which catchment management is capable of influencing DOM concentrations at the point of abstraction,

field studies rarely extending beyond sub-catchment or stream scale." . . .needs rewording – the first part makes sense but the second half is a fragment. Given the lack of evidence is discussed to establish the link between management and response, including something like "research priorities were therefore established" would be logical, rather than ending on the result that evidence is insufficient. Further, the last sentence and second-to-last sentence seem to contradict each other. One says insufficient for wide spread application and then it says the measures have good potential. I think these two sentences could benefit from some rewording to avoid confusion, and make the outcome of the paper more clear. Reword abstract to fit above comments: 80% of the large reservoirs constructed for public water supply in the UK are in upland areas (CIWEM 2011), the majority of which are situated in catchments that contain at least some organic rich soils. Organic matter leaching from these soils imparts a brownish colour to water, primarily due to the presence of dissolved organic matter (DOM). Water companies must ensure DOM concentrations are at negligible concentrations prior to chemical disinfection to prevent the formation of potentially harmful disinfection by-products, and to minimise taste and odour problems. In recent years, water companies have increasingly considered the capacity for catchment interventions to improve raw water quality at source, relieving the need for costly and complex engineering solutions in treatment works, but there is considerable uncertainty around the effectiveness of these measures. The primary evidence gap is the extent to which catchment management is capable of influencing DOM concentrations and treatability at the point of abstraction, as the majority of published field studies have been carried out at plot, sub-catchment or stream reach. Published evidence for the effectiveness of the four main catchment management options utilised on organic soils (ditch blocking, revegetation, reducing forest cover and cessation of managed burning) for reducing DOM concentrations or increasing treatability is generally insufficient to support wider scale application at present. The evidence suggests that the presence of plantation forestry on peat soils is increasing DOM concentrations, though studies assessing the removal of plantation forestry have found that this effect is not rapidly reversed. Although not

widely studied, the available evidence suggests that Sphagnum mosses produce DOM that is more easily removed via conventional treatment processes compared to vascular plants such as heather and grass species. One evidence gap that became apparent during the review process was the extent to which in-reservoir processes may mitigate or exacerbate changes in inflowing DOM when assessed in conjunction with catchment management work. Research priorities to assess these evidence gaps include: experiments to monitor the whole catchment impact of catchment management interventions and to seek to understand the potential interactions between catchment management and within reservoir processes.

2.2: Introduction The opening sentence seeks to make the claim DOC is increasing from 1980- to present, but has a citation that is 1989 and 2001. The rest of the paragraph could be polished in terms of wording. Second sentence repeats the claim of 1980's beginning. Last sentence suggests a more recent reference which is good, but comes after a sentence related to international evidence. Overall, I the paragraph is somewhat awkward, and could more clearly make the argument that a) UK rivers have experienced rising DOC from 1980 to present, and b) similar trends have also been seen elsewhere in the world. The paragraph also doesn't give much quantitative evidence of how much things have changed. It is probably beneficial to include here a figure (or reproduce a figure?) allowing readers to see an example of what this increasing trend looks like.

We suggest including the attached figure showing the annual mean and standard error DOC concentrations from the Upland Waters Monitoring Network sites and rewording the paragraph as follows: Dissolved organic matter (DOM) concentrations in UK surface waters have been rising since the 1980's (e.g. Naden and McDonald, 1989;Watts et al., 2001, Harriman et al., 2001;Robson and Neal, 1996, Freeman et al., 2001;Worrall et al., 2004) and subsequently this has emerged as an international phenomenon affecting surface waters across large areas of Europe and North America (Monteith et al., 2007). The UK Upland Waters Monitoring Network demonstrates that the increase in dissolved organic carbon (DOC) concentrations is sustained, though the rate of increase has reduced in recent years, with DOC concentrations in 2015 being approximately double those seen in the late 1980's (Figure above). At the sub-catchment scale Chapman et al (2010) found that water colour increased by between 22 and 155 percent over a 20 year period between 1986 and 2006.

2.3: Lines 48-60 are describing DOM but no references support the statements. As mentioned in reply to reviewer 1 at comment 1.1.

2.4: Section 1.2 heading issue Line 73 – why is E coli mentioned? Is it related to DOM? If not provide context. Does this need Paragraph 1 would sit better after the DBP/THM paragraph, before the paragraph starting as "Higher concentrations . . ." The reference to the E.coli limit is because elevated DOM in treated water can provide a substrate for bacterial regrowth in the water pipes. Also, the reaction between free chlorine residuals and DOC that produces THMs can reduce the level of chlorine residual to a point where bacterial growth can occur should there be any incursion. As I came at this section from a water industry perspective perhaps this section would be better reframed as these are the potential risks from elevated DOM in water supplies and these are the regulatory measures in place to ensure public safety, which is I think the order the reviewer suggests above, and we will modify the order as suggested.

2.5: Line 91 – Opening line would benefit from a reference The information regarding potential increased treatment processes (lines 91) has resulted from numerous discussions with water industry representatives as part of the NERC funded Freedom project that funded this review. The outcomes of these discussions are currently in the final editing process before publication as a series of briefing notes aimed at the UK water industry so we would suggest referencing Pickard et al 2021 (full reference: Pickard, A.E., Chapman, P.J., Williamson, J., Spears, B.M., Banks, J., Bullen, C., Leith, F., Gaston, L., Moody, C., and Monteith, D.: Rising concentrations of dissolved organic matter in drinking water supplies: can peatland restoration help? FREEDOM-BCCR briefing note I to the water industry. UKRI SPF UK Climate Resilience programme – Project

no. NE/S016937/2. 2021.)

2.6: Section 1.3 is interesting. I think it could be refined to more clearly point out that there are drivers associated with (geo)chemical changes and those associated with hydrological changes; currently they are slightly intertwined. The section could end by summarising the research unknowns that remain. The climate change sentence at the end seems to be less relevant to this section since it is explaining the past and I don't think the single sentence does this issue justice. Maybe this would be better moved to the start of paragraph 2; something along the lines of "As sulphur deposition declines towards pre-industrial levels, hydrological drivers are likely to become the dominant driver of DOM change. Future climate change is predicted to result in increased temperatures and more extreme storm events (Met Office 2019)..." into paragraph 2? Since the early 2000's a number of hypotheses have been advanced to explain regional scale increases in DOM concentrations, some focussing on the impact of land use change, others focussing on the impact of climate change (on DOC production and hydrological processes) and others on the interaction between nutrients within the peat'. These initially focussed on increasing temperatures (e.g. Freeman et al., 2001), drought-rewet cycles (e.g. Watts et al., 2001), and increasing atmospheric carbon dioxide concentrations (Freeman et al., 2004), before negative correlations with indicators of acid deposition, such as sulphate concentration, became increasingly apparent in the USA (Stoddard et al., 2003) and UK (Evans et al., 2005). A regional study of DOC trends (Monteith et al., 2007) demonstrated consistent significant negative relationships between rates of change in acid anion concentrations (sulphate & chloride) and rates of change in DOC, and that the effect was more marked for waters with lower concentrations of calcium and magnesium (i.e. base cations). Hence, sites with soils that were least able to buffer the effects of deposited acidity were the most responsive. The links with changes in atmospheric deposition have since been supported by studies of soil cores (Clark et al., 2011) and field experiments (Evans et al., 2012;Ekström et al., 2015). Hruska et al. (2009) demonstrated that ionic strength (a measure of the electric charge produced by ions in water) is a particularly effective

chemical predictor of change in DOC. A reduction in the deposition of acid anions from the atmosphere reduces both the acidity of soil and the ionic strength of soil water, and together these processes appear to increase the solubility of soil organic matter and hence the increasing concentration of DOM draining from organic rich upland soils. DOM concentration in soil solution and surface waters is also known to respond positively to variation in temperature (e.g. Vance and David, 1991), while shifts from vertical to more lateral routing of flow paths during periods of heavy rain have also been found to increase concentrations in some circumstances (e.g. Austnes et al., 2010), with increases in DOC concentration being primarily driven by the increase in water table at the event scale (Rosset et al., 2019). Shifts in stream DOC character, and hence treatability, are also possible following changes in flow path routing as a result of DOC inputs from different source pools (Hood et al., 2006). Long-term increases in DOC in southern Sweden have been linked to the combination of decreasing sulphate deposition and a multi-decadal increase in precipitation and consequently river flow (Erlandsson et al., 2008). Future climate change, particularly in relation to increasing temperatures and a change in total rainfall and an increase in the intensity of storm events (Met Office, 2019), is therefore likely to influence future DOM trajectories and has the potential to become the dominant driver as atmospheric pollutant deposition declines toward pre-industrial levels.

2.7: Line 140 – I think a sentence that is pivoting like this needs to being with Also, or Further or In addition. Perhaps "However" would be a good start to the sentence, and would show the direction of the sentence more clearly.

2.8: Line 142 – It is asserted here that catchment management activities are not seen at the point of abstraction (presumably you mean at the reservoir outlet?) but it is not obvious from the prior text this is established. Is there a published paper saying this, or just a "hunch"? Section 2 obviously goes into this, but in this case the text is out of order. We would suggest rewriting this to make it clearer we're referring to absence of published evidence of measured effects at the point of abstraction (which is usually,

but not always, the reservoir outlet), not evidence of absence. This is an area the water industry in the UK is picking up on but we are not aware of any published studies to date that have demonstrated the effectiveness (or lack) of catchment management at reducing DOC concentrations at the point of abstraction.

2.9: Line 147 – ditch blocking mentioned here but not above in the catchment management section, so seems out of context. I find the aims statement buried in sub-section 1.4, quite deep into the manuscript, to be somewhat awkward. The aims statement is weak in that the aim of the paper "bring together information" and "contribute to our understanding" and "go on to examine". These aims lack specificity and are overly general in my view, making it difficult for the reader to clearly understand what the outcomes of the paper will be. Whilst I acknowledge it is a review paper, a good review can still have specific aims. E.g "Is there evidence that . . .". or "The review is used to develop a conceptual model. . .." As suggested for reviewer 1, an opening paragraph to the introduction stating the aims more strongly – the question being is there evidence to support peatland restoration through catchment management being used as a tool to improve water quality at source by water companies in the UK? – would help. I think it may also help the information flow more easily – this is the aim of the review, this is the background as to why DOC in raw water has become an issue, and these are the catchment management tools that have been trialled to help solve the issue. Then the in-lake processing becomes part of an unknown within the context of catchment management – we don't know what may change within the lakes so it may be that in-lake processes "deal with" most of the extra DOC coming in anyway so reducing inflowing DOC doesn't reduce outflowing concentrations to the same extent.

2.10: Line 155 – this paragraph flows well Thanks.

2.11: Line 166 – the line "While these results are persuasive, they do not necessarily imply that effects will be translated through to surface waters and ultimately to the point of abstraction" seems unnecessary at this point, between describing pore water changes and ditch water changes. We will remove this sentence

2.12: Line 175 – I struggled to follow this logic. If a study was from a hydrological point of view then there is a view that DOC decreases? But Wilson says that DOC load went down but not conc? This section could benefit from the authors make a conceptual diagram to synthesise the results. The aim of this was to show that DOC load only decreased because water flux decreased as concentrations stayed the same. As water inputs to the catchment (rainfall) are highly unlikely to have declined by ∼90% since the start and although peat surface can move in response to rainfall inputs they don't have any evidence to suggest it's done so by the volume required, then it is likely that the water is still leaving the catchment but via an unmonitored route such as overland flow so fluxes are higher than they are reporting.

2.13: Line 207 – I don't think the last two sentences of this paragraph are relevant for an international journal. We will remove these sentences.

2.14: Line 331 – This sentence is not really adding anything – It is great people are doing more work, but in this paper it would be just better to present published findings. We will remove this sentence.

2.15: Section 3 – this section reads reasonably well. The key is whether the creation or consumption of DOM is big or small relative to the a) the observed increases mentioned in the introduction, and b) the catchment management activities. I cant tell from reading this. We will include this as one of the uncertainties in our understanding of the interactions between catchment management and inn-lake processes.

2.16: Conclusions – A lot of the conclusions seems more like opinion, and I'm looking for more specific summary here – scientifically what is the evidence, per km2 of land, that DOC will go up or down for a given intervention? Is one intervention more effective? How does land management actions compare to in-lake processes in potential amounts of DOC removal? Finally, is there a role for models to help compute a DOM budget? It seems that modelling is overlooked, but can be useful for assessing this issue and so should be in the review.

We feel that part of the conclusion of this review is that the evidence is not yet there to be able to put a figure on a given intervention changing DOC concentrations by X per km2 of land under differing management. The aim at the start of the project was to be able to do this but we haven't found sufficient evidence in the published literature. There is a role for modelling but feel that this is outside the scope of this review. We can include mention of models in the conclusions section as part of the suite of tools available to assess this issue but it is possible that parameterising such models would have the same issue of data availability. We agree that a good future point to reach would be if a water company (or other interested party) could have the information that they are blocking ditches on X ha of a catchment, revegetating Y ha and restoring Z ha of conifer plantation to bog and use that to predict a reduction in DOC by . . .mg/l but there isn't this level of information available that we can identify at present.
* * *
[Figure]

**Fig. 1.**